# The Roles of Primary Cilia in Cardiovascular Diseases

**DOI:** 10.3390/cells7120233

**Published:** 2018-11-27

**Authors:** Rajasekharreddy Pala, Maha Jamal, Qamar Alshammari, Surya M. Nauli

**Affiliations:** 1Department of Biomedical & Pharmaceutical Sciences, Harry and Diane Rinker Health Science Campus, Chapman University, 9401 Jeronimo Road, Irvine, CA 92618-1908, USA; rrpala@chapman.edu (R.P.); jamal106@mail.chapman.edu (M.J.); qalshammari@chapman.edu (Q.A.); 2Department of Medicine, University of California Irvine, Irvine, CA 92868, USA

**Keywords:** primary cilia, calcium, nitric oxide, biochemical signaling, hypertension, aneurysm, atherosclerosis

## Abstract

Primary cilia are microtubule-based organelles found in most mammalian cell types. Cilia act as sensory organelles that transmit extracellular clues into intracellular signals for molecular and cellular responses. Biochemical and molecular defects in primary cilia are associated with a wide range of diseases, termed ciliopathies, with phenotypes ranging from polycystic kidney disease, liver disorders, mental retardation, and obesity to cardiovascular diseases. Primary cilia in vascular endothelia protrude into the lumen of blood vessels and function as molecular switches for calcium (Ca^2+^) and nitric oxide (NO) signaling. As mechanosensory organelles, endothelial cilia are involved in blood flow sensing. Dysfunction in endothelial cilia contributes to aberrant fluid-sensing and thus results in vascular disorders, including hypertension, aneurysm, and atherosclerosis. This review focuses on the most recent findings on the roles of endothelial primary cilia within vascular biology and alludes to the possibility of primary cilium as a therapeutic target for cardiovascular disorders.

## 1. Introduction

Cilia have been studied for their motile functions of lung epithelium, sperm cells, as well as in other organisms (such as algae) [1]. These motile cilia, in most cases, have 9 + 2 microtubule structural arrangement. Researchers have also gained interest in studying non-motile cilia [2]. Non-motile cilia also known as primary cilia have a 9 + 0 microtubule structural arrangement. Defects in primary cilia could cause various life-threatening diseases in humans, such as neural tube defects, which result in numerous abnormalities of the brain and spinal cord in patients diagnosed with Meckel syndrome, for example [3]. Several studies have shown that primary cilia are recognized as mechanical and chemical sensory organelles which serve as antennae to transmit extracellular to intracellular signaling mechanisms. Because primary cilia act as sensory organelles by which cells sense and transduce extracellular signals [4], any defects in primary cilia function could potentially cause several diseases which are collectively known as ‘ciliopathies’ (Table 1). The list of human ciliopathies has increased in recent years [5,6]. Mutations in approximately 50 genes have revealed to alter ciliary assembly or function, and as many as 1000 different ciliary proteins are still with undetermined functions and required further investigation. Hence, abnormal ciliary proteins can be associated with a single organ dysfunction to systemic multiple organ complications depending on the type of cells affected [7].

Cilia dysfunction has been implicated in polycystic kidney disease (PKD), obesity, nephronophthisis, mental retardation, Bardet-Biedl syndrome, oral facial syndrome, vascular diseases and others [7]. Specifically, impaired primary cilia on endothelial cells have important clinical consequences and are associated with many vascular diseases [8,9]. Although it has been over a century since primary cilia have been visualized, the study of their mechano- and chemo-sensory roles remains relatively a new field of study. Additionally, studies on endothelial primary cilia function of the vascular system and primary cilia as a therapeutic target for vascular diseases are still limited. In this review, the roles of primary cilia will be discussed with emphasis on the cardiovascular diseases [10,11,12,13]. Hence, it is important to have a clear understanding about ciliary structure and functional ciliary proteins to investigate how cilia dysfunction can contribute to vascular disorders of hypertension, aneurysm and atherosclerosis.

## 2. Cilia Structure

A cilium is considered as a cellular organelle, which is primarily composed of a membrane, soluble compartment, axoneme, basal body, and ciliary tip [38]. A cilium extends from a basal body complex, which is mainly composed of two centrioles. One of the centrioles is known as the mother centriole, to which the ciliary axoneme is ingrained beneath the cell membrane. The cilium structure contains the microtubular portions of cytoskeletal core unit called the axoneme (Figure 1). The axonemal structure contains nine peripheral doublet microtubules, which are made of alpha- and beta-tubulins and are post-translationally acetylated to support the long cilia structure [39,40,41]. The non-motile axoneme structure lacks the central pair of microtubules (9 + 0) [42,43]. In the blastocyst nodal cilia, the axoneme lacks the central pair of microtubules (9 + 0) but shows motility. This exception requires both dynein arms for motility [44]. Of note is that the lack of radial spokes induces rotational motion instead of beating motion, suggesting that the absence of radial spokes allows nodal cilia to rotate unidirectionally but, as a trade-off, renders them ultrastructurally fragile [44]. While in most cases of motile cilia, the axoneme contains nine peripheral doublet microtubules and a central pair of microtubules (9 + 2) [45]. There is a connection between the microtubular portion of the cytoskeleton and the ciliary axoneme, and the disruption of cytoplasmic microtubules or actin filaments, which could affect microtubules assembly resulting in the loss of ciliary structural integrity and mechanosensory function [46,47,48]. On the other hand, protein entering and exiting through cilium is controlled by a proteomic barrier at the ciliary base that encompasses a transition zone which separates cytosol from the cilia [49,50,51,52]. The ciliary membrane is connected with the plasma membrane but possesses a lipid bilayer composition that differs from the plasma membrane compositions [53]. The periciliary membrane, also known as the transition-membrane, connects ciliary and plasma membrane to form the ciliary pocket [54,55]. In addition to its fundamental structural role, the basal body connected to the transition-membrane is thought to regulate protein entry and exit from the ciliary compartment. Furthermore, the mechanoreceptors, protein transporters, sensory proteins and ligand-gated ion channels are involved in signal transduction enclosed within the ciliary membrane. The axoneme allows the intraflagellar signaling and intraflagellar transport (IFT) activities along with ciliary shaft using the soluble compartment, also called cilioplasm. Primary cilium lacking ribosomes is incapable of producing its own proteins required for the elongation and continuous turnover of axoneme necessary for self-safeguarding. Moreover, each part of this cilia structure is crucial to support various signaling molecules. Some of the more established cilia-dependent signaling pathways are already described [38].

## 3. Primary Cilia as a Blood Flow Sensor

Flow sensing by the cilia permit cells to sense blood flow along the blood vessels, urine flow through kidneys, bile acid in the liver, pancreatic secretions in the duodenum, nodal flow in Hansen’s node (the site which determines the patterns the anterior-posterior axis of the embryo during gastrulation), interstitial fluid flow within the bone canaliculi, and potentially other systems/organs [56]. The function of the vascular system depends on the mechanical fluid flow signaling from the blood flow. Several studies have also reported that the presence of primary cilia in major circulatory systems including endocardia [13,57], arteries [58,59], veins [60], corneal endothelium [61,62], and smooth muscle cells of both arterial and airway endothelia [63,64]. The continuous contraction and relaxation of smooth muscle cells produce changes in the blood vessels diameter, which is important for normal blood flow [65,66,67]. Increase in vascular stiffness is a major cause of hypertension, which leads to complications including ventricular hypertrophy, vascular aneurysm and atherosclerosis [68,69,70,71,72]. These changes suggest that smooth muscle cells or neuronal regulations are important in regulating the vascular tone in addition to the mechanical fluid-flow within the blood vessel. The regulation of circulatory function is acquired by neuronal regulators through central and/or peripheral neurons [73,74,75].

The mechanical fluid-flow provides local regulation or autoregulation within a blood vessel. For example, autoregulation is required to achieve immediate blood flow control in specified area of the tissue. Autoregulation is independent of the neighboring tissues and has little to no effect on the surrounding tissues [76,77]. In an isolated blood vessel, the sudden increase of transmural blood pressure causes a reduced vessel diameter [78,79,80], whereas high flow stress increases vessel diameter [67,80,81,82]. As such, the lining of the inner surface of vascular blood vessels are endothelial cells with primary cilia protrusions, which can sense changes in blood velocity and pressure and convert these mechanical signals into changes of vascular smooth muscle tone [83,84]. In a biophysical perspective, fluid-shear stress refers to the partial or frictional force of blood flow as it brushes against the vascular endothelia [85]. This frictional force is not stable because blood flow changes with each heart muscle contraction, resulting in pulsatile patterns of blood flow [85]. As a result, blood flow through a vessel creates different types of forces such as stretch, compression, cyclic strain, pressure and shear stress. While these forces may be practically impossible to differentiate *in vivo*, they can be independently studied in *in vitro* and *ex vivo* studies [86].

Our earlier studies show that the primary endothelial cilia act as a mechanosensor in *in vitro* (mouse aortic endothelial cells), *ex vivo* (isolated mouse arteries, blood vessels from human patients) and *in vivo* (mouse models) [11,12,58]. Ciliary length is also positively correlated with mechanosensory action. Blood vessels with relatively a low fluid force have longer cilia while blood vessels with a high fluid force are devoid of cilia or have very short cilia. In addition, the changes in fluid dynamics affect endothelial cilia distribution and depend on fluid-flow intensity with longer cilia present in lower fluid-flow areas [13,87]. This is because of the inability of primary cilia to stand against high levels of fluid flow, which results in ciliary disassembly and loss of intraflagellar transport which is necessary for ciliary reassembly [88]. Subsequently, the mechanosensing function of cilia in high fluid flow areas could be replaced by other mechanisms like glycocalyx to sense higher shear forces [89].

Primary cilia have a critical role in sensing the extracellular stimuli, such as odorant or chemical (chemosensory) and movement (mechanosensory). These stimulations are then translated into intracellular signals. As a mechanosensor, a primary cilium can sense the fluid-flow in multiple cell types including renal epithelial and vascular endothelial cells [12,18,90]. Polycystin-1 (PC1) and polycystin-2 (PC2) form a mechanosensory complex in the primary cilia. It is recently shown that the PC1 and PC2 form a complex and are assembled in a stoichiometry of 3 PC2 for every PC1 molecule [91]. The PC1 and PC2 complex detects the bending of the cilia by the fluid flow leading to an increase in Ca^2+^ influx and an inhibition of the regulated intramembrane proteolysis (RIP) of PC1 by keeping the signal transducer and activator of transcription (STAT) factor 6 and its coactivator P100 in a complex bound to PC1 tail [92,93]. This is how primary cilia is thought to promote proliferation and differentiation through fluid-shear stress. On the other hand, the absence or lack of flow as well as loss or dysfunction of cilia, PC1, or PC2 decrease Ca^2+^ influx and activate RIP that allows STAT6 and P100 to translocate to the nucleus and stimulate transcription resulting in uncontrolled cell proliferation and cyst formation [94,95]. In particular, PC1 and PC2 are widely expressed across the vasculature, and they are hypothesized to play a major role in the development, maintenance, and function of the myoelastic arteries [96,97,98]. These observations indicate a direct pathogenic role for both PC1 and PC2 in the vascular complications of hypertension, aneurysm and/or atherosclerosis.

## 4. Role of Primary Cilia in Heart Development

Nodal cilia probably have the earliest cilia function during embryonic development. During gastrulation period, both motile (nodal) and non-motile cilia at the embryonic node play an important role in regulating signaling cascades required for the formation of left-right asymmetry, a process which regulates the early stages of cardiogenesis and connection to the blood vessels [15,99,100,101,102]. Fluid flow plays an important role in trabeculation, cardiac cell proliferation, and formation of conduction system, in addition to changes in fluid-shear forces, which lead to cardiac diseases. Cilia in cardiomyocytes have a series of receptors, which take part in regulating cellular signaling mechanisms required for the continuous differentiation, morphogenesis and development of the heart [103,104,105,106]. Independent studies have established the important role of heart cilia in cardiac development. Defects in cilia structure or function lead to severe inherited cardiac diseases. Also, defects in cilia result in a variety of heart developmental defects such as arterial and ventricular septum defects [107,108], abnormal looping, and remodeling of the heart tube into a multi-chambered organ [109,110,111,112,113,114] or myocardial wall disorganization [115]. Moreover, mice with a mutation in cilia structural gene *ift88*, *kif3a* or *kif3b* are characterized by severe heart phenotypes including hypoplasia of the endocardial cushions, a reduction in ventricular trabeculation, and an increase in volume of pericardial space including defective cardiac looping [102]. A variety of signaling pathways are involved directly or indirectly in heart development. For example, Hedgehog (Hh) signaling coordinated by primary cilia in a variety of cells controls tissue patterning and promotes the activation of different transcriptional factors involved in different cellular signaling mechanisms during homeostasis in vertebrates [116,117]. As a result, defects in primary cilia Hh signaling leads to severe cardiac disorders including congenital heart diseases [118]. Another example of a signaling pathway which plays an important role in cardiac morphogenesis is the superfamily of Transforming Growth Factorβ/Bone Morphogenic Protein (TGFβ/BMP). TGFβ/BMP signaling network is involved in a wide range of cellular mechanisms and processes and is therefore fundamentally vital during tissue homeostasis and morphogenesis [119]. Recent studies show that primary cilia can regulate the canonical TGFβ signaling network through the activation of transcription factors Smad2/3 at the ciliary pocket [104]. Furthermore, the TGFβ ligand, TGF-β1, stimulates the differentiation of stem cells into cardiomyocytes and that *Ift88/Tg737* (*Tg737^orpk^*) mouse embryonic fibroblasts are characterized by decreased TGFβ activity associated with reduced clathrin-dependent endocytosis activity at the ciliary base, suggesting that cardiac primary cilia play a direct role in regulating TGFβ signaling during cardiomyogenesis. Recent findings further show that platelet-derived growth factor receptor-α (PDGFRα) localizes to primary cilia in mutant mouse heart, indicating that a portion of the PDGF signaling pathway is associated with cardiac primary cilia during cardiac morphogenesis and development [105]. The localization of PDGFRα causes downregulation of Hh signaling in primary cilia and causes diminished ventricular wall thickness and ventricular septal defect [105]. Further, mice studies show that mutated or the absence of PDGFRα, consequences arise in prenatal mortality such as heart defects including weakened myocardium, thinned septa and valve, outflow tract, and aortic branch malformations [120,121,122]. Taken together, the PDGF signaling system might be specifically coordinated by cardiac primary cilia, potentially acting as signaling hubs facilitating the cross-talk between different signaling networks in order to coordinate cardiogenesis.

## 5. Role of Primary Cilia in Biochemical Signaling and Hypertension

As mechanosensory organelles, primary cilia depend on various receptors expressed on the ciliary membrane. Vascular endothelial cells lining the blood vessel wall are in continuous contact with blood flow forces. Activation of primary cilia by blood flow leads to the activation of PC1 and PC2 resulting in an intracellular Ca^2+^ signaling network involving calmodulin (CaM), calcium-dependent protein kinase (PKC), serine-threonine kinase/protein kinase B (Akt/PKB) and endothelial nitric oxide synthase (eNOS). Such biochemical reaction generates nitric oxide (NO) leading to vasodilation (Figure 2). There are two major proposed mechanisms for primary cilia detection of blood flow forces [85]. The first suggests that ciliary bending occurs upon exposure to blood flow-pressure force, which triggers cytoskeletal distortion. The second suggests that cilia bending triggers activation of PC1 mechanosensory protein and PC2 cation Ca^2+^ channels. It is proposed that the increase in intracellular Ca^2+^ is caused by an increase in intraciliary Ca^2+^ [123], whereas another study has suggested that Ca^2+^ could be rallied in both directions between the cilia and the cytoplasm [124]. While differences in the intraciliary Ca^2+^ can be due to the sensitivity of the cilia-specific Ca^2+^ probes [125], both studies show a consensus that mechanosensing function of cilium involves cytosolic Ca^2+^ signaling as shown independently by other laboratories [17,126,127]. Thus, it is fair to assess that primary cilia are Ca^2+^-responsive mechanosensors that can trigger a diverse biochemical signaling.

Regardless, the cytosolic Ca^2+^ forms complexes with CaM, and the Ca^2+^-CaM complex has been shown to indirectly activate eNOS through activation of the AKT/PKB signaling which activates AMPK, a known stimulator of eNOS [128]. Inhibition of Ca^2+^-dependent PKC, Akt/PKB, or CaM activity downstream of Ca^2+^ signaling have no effect on the flow induced intracellular Ca^2+^ increase, although there is a loss of NO synthesis [11]. This indicates that the Ca^2+^ signaling is upstream of the biochemical reaction in producing NO. Though eNOS triggering is principally a Ca^2+^-dependent process, some studies have suggested a Ca^2+^-independent pathway in NO biosynthesis is also possible. This Ca^2+^-independent pathway depends on the heat shock protein 90 (HSP90) [129,130]. HSP90 is a molecular chaperone, but it may also act as a signal transduction agent concomitant with eNOS in several systems, including the cardiovascular system. HSP90 also localizes to primary cilia [131]. Although its activation can increase eNOS action in presence of Ca^2+^-CaM [129,132,133], it is unclear if cytosolic HSP90 is involved in this signaling pathway.

Dopamine signaling is considered to be an important signaling mechanism in the nervous, immune, cardiovascular, and renal systems [134]. Dopamine is an endogenous catecholamine hormone that is mainly produced in the brain and adrenal gland and is also biosynthesized in renal proximal tubules [135,136,137]. Dopamine, an endogenous hormone in the sympathetic nervous system, is known to be intricated in the regulation of hypertension. For example, abnormalities in dopamine signaling can contribute to high blood pressure in humans. The five G-protein-couple dopamine receptors (DR) are categorized into D1-like (DR1 and DR5) and D2-like (DR2, DR3, and DR4) families. Several *in vitro* and *in vivo* experiments confirm the presence of Dopamine 1-like receptors, DR1 and DR5, on primary cilia [59,138,139,140,141]. Studies have identified DR5 receptors in cultured mouse vascular endothelial cilia and mouse arteries *in vivo*. The DR modulates cilia mechanosensory function by altering fluid flow sensitivity. Rat studies also show that dis-integrin and metalloproteinase with thrombospondin motifs 16 (*Adamts16*) play a crucial role in blood pressure control. Further, interruption of the *Adamts16* gene results in longer vascular endothelial primary cilia and significantly lower systolic blood pressure [58]. To date, there are no drugs available that specifically target DR in the cilia, but studies using agents selective for DR1-like receptor subtypes have shown vasodilatory outcomes in peripheral arteries. Activation of DR5 using dopamine increases ciliary length while inhibition of DR5 leads to the loss of ciliary sensory (chemo and mechano) activity [59]. These results are confirmed by challenging endothelial ciliary knockout cells, *Pkd1*^−/−^ and no or short cilia *Tg737^orpk/orpk^* with dopamine under static conditions, resulted in a considerably less Ca^2+^ influx than wild-type endothelial cells. As Ca^2+^ fluxes in these cells are often concomitant with activation of eNOS, the results may indicate a potential reestablishment of the missing vasodilatory reactions caused by a failed ciliary generation of NO biosynthesis. Likewise, there are DR within blood vessels in human, and activation of DR triggers a vasodilatory action [142].

Cilia dysfunction causes abnormal Ca^2+^ signaling and kidney disorders such as autosomal dominant polycystic kidney disease (ADPKD), which is a genetic disease caused by a mutation in ciliary PC1 or PC2 [6]. Cardiovascular malformations including high blood pressure and left ventricular hypertrophy notably contribute to mortality in ADPKD patients. A recent clinical review involving 1877 ADPKD patients shows that the use of antihypertensive medications in ADPKD patients have been increased from 32% in 1991 to 62% in 2008 [143]. This has important clinical consequences as another study has found that border-line hypertension in ADPKD patients show a better response with a dopamine precursor relatively to the angiotensin-converting enzyme inhibitor [144]. When individuals are perfused with 0.25–0.5 μg/kg/min of dopamine, the results indicated an upward trend in flow-mediated dilation in ADPKD patients and reported a statistically significant decrease in hypertension [145]. It is currently studied to better understand if the dopamine-induced vasodilation is a cilia-dependent process [146]. A more recent study, however, seems to support the idea of cilia involvement in hypertension [147]. The study shows that cilia function is impaired in endothelial cells from patients with pulmonary arterial hypertension due to the inflammation, and cilia length plays an important role in response to inflammatory signaling, such as pro-inflammatory cytokines and/or anti-inflammatory interleukins. The results show that the pro-inflammatory cytokines help in increase cilia length and is PKA/PKC-dependent, whereas anti-inflammatory interleukins induce a reverse effect on cilia length. It is therefore postulated that the length of endothelial cilia is associated with endothelial function and pulmonary arterial pressure.

## 6. Role of Primary Cilia in Vascular Aneurysm

An aneurysm is a formation of an abnormal swelling in a weak area of a blood vessel that can rupture, leading to bleeding and possibly to death. The most common arteries that can be affected by aneurysm are cerebral arteries and aortic artery. Aneurysm formation and rupture are considered one of the major complications associated with ADPKD, in which PC1 is required for structural integrity of blood vessel [148]. Thus, PC1 and PC2 functions are required in blood vessels [97,98,149], and, any abnormalities in either protein leads to aneurysm formation [150]. Of note: In ADPKD patients, the aneurysm can occur in different arteries such as the aorta, splenic, coronary, and cerebral arteries [151,152,153,154].

Within the arteries, primary cilia play an important role in the structure and the function of endothelial cells [12,60]. Therefore, the absence or dysfunction of primary cilia can induce aneurysm formation and progression during vascular injuries [10,155]. Vascular aneurysms are associated with tissue remodeling due to unusual proliferation of the endothelial cell layers through the hemodynamic fluctuations in fluid-shear forces [156]. Endothelial cilia are required for shear stress-induced Ca^2+^ influx and NO signaling [11], and eNOS deficiency is the hallmark of endothelial dysfunction and associated with cardiovascular complications including aneurysm, indicating the protective role of eNOS [157]. Primary cilia regulate endothelial actin organization and focal adhesion assembly that can affect directional migration and cell permeability through hsp27 and Notch/foxc1b signaling [158,159]. It is therefore thought that the mechano-sensation of primary cilia is essential in promoting proper vascular development.

Previously, we showed that the similarity of the pathogenesis between cyst formation and aneurysm associated with PKD in mice models (*PdgfβCre:Survivin^flox/flox^*, *PdgfβCre:Pkd1^flox/flox^* and *PdgfβCre:Tg737^flox/flox^*). Dysfunction of the primary cilia induces an abnormal survivin expression that results in irregular cytokinesis leading to cell polyploidy, multi-mitotic spindle formation and aberrant cell division orientation. This abnormality in symmetrical cell division and cell ploidy leads to the extension of tissue architecture, developing cysts in the kidney and aneurysm in the vasculature [10]. PKC and Akt are downstream signaling messengers of primary cilia, and they regulate survivin expression following primary cilia activation. Akt is downstream of PKC and can regulate Nuclear Factor-κB, which regulates the expression of survivin. All in all, the inability of primary endothelial cilia to respond to fluid flow can contribute to the vascular aneurysm.

## 7. Role of Primary Cilia in Atherosclerosis

Atherosclerosis plaques mainly develop in the arterial system with bifurcations, branch points, or the inner curvature of arched arteries. Atherosclerosis plaques are often observed at sites with low and oscillating fluid-flow within the embryonic cardiovascular system [13,57]. Plaques happen most frequently in areas of great curvature and branch points in addition to low fluid forces or non-unidirectional flow [160,161]. Like cilia which are present only at the regions of inner curvature of the artery arch [13], atherosclerotic plaques do not happen homogenously along the circulatory system. A recent report confirms that removing endothelial cilia from the vascular branch points causes abnormal fluid-flow responses that contribute to the atherosclerosis [162]. Moreover, exposure of endothelial cells to oscillatory fluid-flow results in the disengagement of eNOS, which promotes reactive oxygen species (ROS) formation rather than NO, leading to atherosclerosis plaque growth [163]. There is an upregulation of inflammatory gene expression in areas with disturbed blood flow, and this further promotes plaque formation and hyperlipidemia [164,165].

The role of primary cilia in the development of atherosclerosis has been revealed in the apolipoprotein-E-deficient mouse model (*Apoe*^−/−^) with a high fat and cholesterol diet [162]. Increasing numbers of the endothelial primary cilia existed in atherogenesis areas under hyperlipidemia-induced lesion formation. *Tek-Cre•Ift88^C/−^•Apoe*^−/−^, in which endothelial *Ift88* was specifically ablated, displayed a significantly greater increase in plaque formation compared to that established by their wildtype littermates. The lack of endothelial cilia in vascular branches result in significant upregulation lymphocyte markers, macrophage marker genes, along with proinflammatory cytokines [162]. Atherosclerosis lesions increase in the mice who lack endothelial cilia by 59% in females, and 67% in males as compared to the control mice. This is measured by counting atherosclerotic lesioned surface area. Furthermore, lacking endothelial cilia enhances inflammatory gene expression and a decrease in endothelial nitric oxide synthase activity. Hence, it is proposed that vascular endothelial cilia play an important role in control of atherosclerosis.

## 8. Role of Primary Cilia in Cell Proliferation

Not only do primary cilia provide a sensory signaling hub, they also play an important role in cell proliferation. Ciliogenesis begins at the G1/G0 phase of the cell cycle, and resorption or disassembly of cilia starts after the cell cycle re-entry. Primary cilia formation is influenced by the coordination of assembly/disassembly equilibrium, IFT system, and membrane trafficking [166]. Specifically, ciliogenesis involves multiple steps and is correlated with cell division. First, the centrosome travels to the cell surface and the basal body is formed by the mother centriole to nucleate ciliary axoneme at the G1/G0 phase of the cell cycle. This step which involves membrane docking is regulated by the distal appendage proteins, such as centrosomal protein 164 (Cep164). On the other hand, CP110, Ofd1, and trichoplein are negative regulators of ciliogenesis targeting ciliary extension. Second, elongation of the cilium and maintenance of ciliary length occur. This process is negatively regulated by Nde1 until mature primary cilium is formed. Third, upon cell cycle entry, ciliary resorption occurs followed by axoneme shortening. Ciliary disassembly is controlled by Aurora A-HDAC6, Nek2-Kif24, and Plk1-Kif2A pathways. Fourth, the basal body is released from cilia; thus, centrioles (centrosome) become free to act as microtubule organizing center (MTOC) or spindle poles during mitosis [166,167,168].

In tumors, cilia are not present on most proliferative cells suggesting that although cilia are not directly required during cell proliferation, they do play a key role in the entry and exit of mitosis [169,170,171]. PC1 has been shown to mediate JAK/STAT pathway [172]. Ciliary PC1 is able to activate STAT3; when the cytoplasmic tail of PC1 is cleaved in response to fluid-flow, it can coactivate STAT-1, 3, and 6 as well as JAK2 [92]. The PC1 tail triggers several cytokines and growth factor signaling, amplifying the cellular response and potentially leading to an increase in L-arginine thus arresting cell proliferation.

Although the reason of the absence of cilia in cancer cells is not exactly known, this phenomenon is arguably not surprising given that the presence of cilia is a cell-cycle-dependent process [173]. Thus, cilia are not expected to be present in highly proliferative cells. However, what complicates the discussion is that primary cilia have also been reported in cancers, including in medulloblastoma [34,174], basal cell [33] and gastroinstestinal stroma cells [175]. A recent study suggests a possibility of an enzymatic effect in cancer cells [176]. It is shown that posttranslational modification of ciliary tubulin is affected and resulted in less robust formation of primary cilia. Lacking proper posttranslational modification in ciliary exoneme may therefore increase a risk factor for cancer development [176].

## 9. Conclusions and Perspective

Both primary cilia structure and sensory functions are essential for normal tissue homeostasis and function. The *in vitro* and *ex vivo* fluid-flow studies have greatly advanced our knowledge of the chemo- and mechano-sensory function of primary cilia in cardiovascular systems. More studies are warranted towards clinical intervention for hypertension, aneurysm and atherosclerosis. Unfortunately, there are no pharmacological agents available that selectively target primary cilia. While this review mostly represents a small portion of possible connections between primary cilia and cardiovascular disorders, we may need a large-scale screening study to include potential pharmacological agents in order to understand whether or not targeting sensory functions of primary cilia would result in better cardiovascular outcomes.

Primary cilia are ubiquitously present in many organ systems, including the cardiovascular system. Emerging data suggest that cilium dysfunction is a primary cause in many cardiac and vascular disorders. Over the past years, researchers have provided tremendous advances in understanding of the basic cellular and molecular functions of primary cilia. Despite the fact that more research is needed, we should also extend ourselves by integrating the basic science knowledge into clinical considerations and perspectives. Otherwise, we are not able to see the forest because we are too focused on the trees.

## Figures and Tables

**Figure 1 cells-07-00233-f001:**
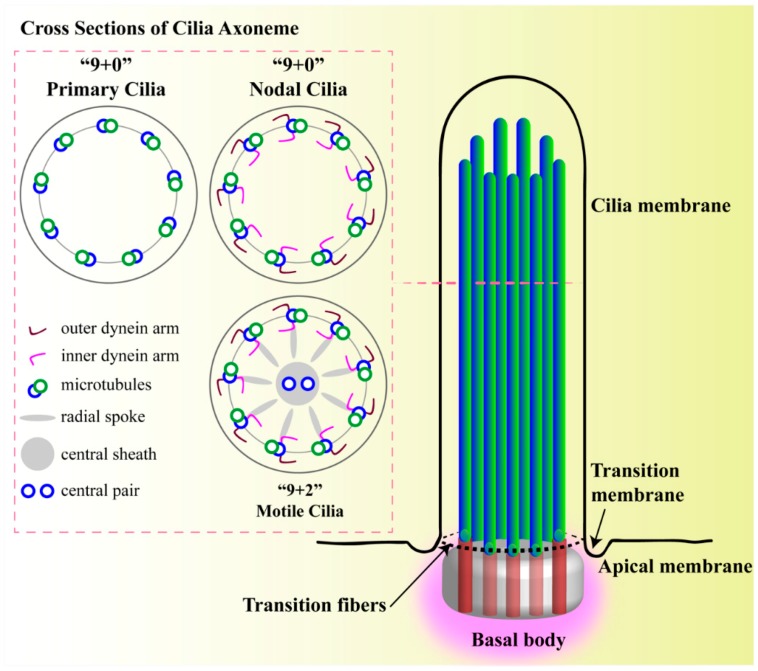
Structure of primary cilium. A cilium is a membrane-bound structure and composed of multiple central pairs of microtubules (axoneme) running from the basal body. A basal body is a microtubule-based structure composed of mother and daughter centrioles. The ciliary membrane and axoneme contributes to the upper part of the cilium. The ciliary membrane is continuous with the cell membrane, but they have their own proteins, ion channels and/or receptors. The ciliary skeleton may have 9 + 0 or 9 + 2 axoneme compositions. Most 9 + 0 cilia lack inner and outer dynein arms, radial spokes, and central sheath and are commonly referred as non-motile primary cilia. Some 9 + 0 cilia lack the central microtubule only and are motile. Between the cell membrane and cilium, there is a transition-membrane at the junction of the basal body acting as a barrier for molecules to enter or exit from the primary cilium.

**Figure 2 cells-07-00233-f002:**
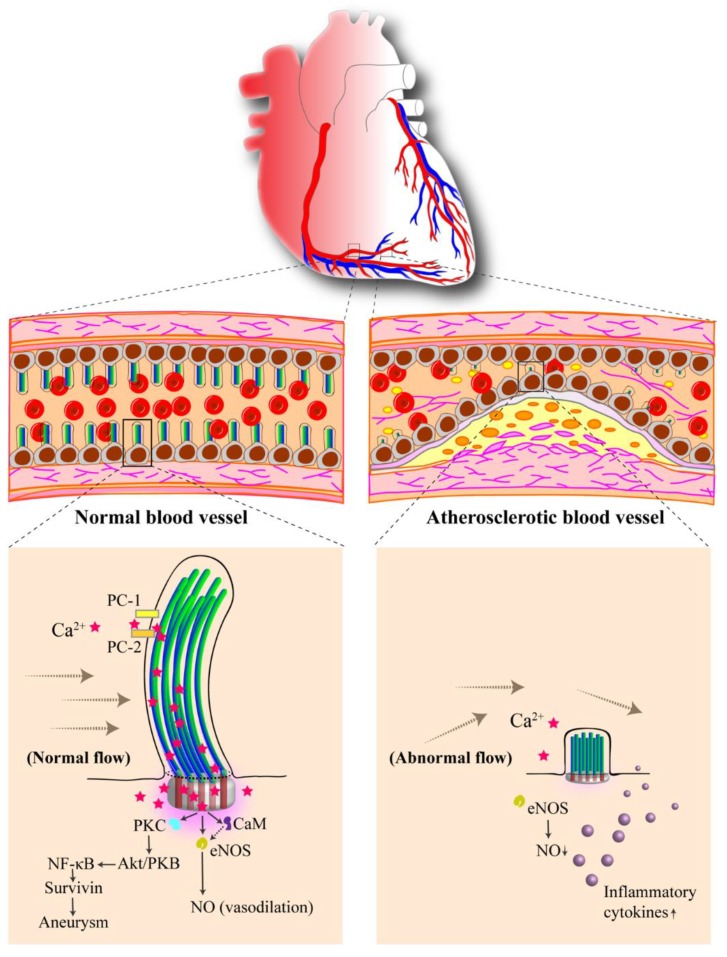
Vascular endothelial cilia sense the blood flow along the blood vessel. Primary cilia are structural compartments that house many mechanosensory proteins. Ciliary bending occurs upon blood-flow stimulation, and polycystin-1 (PC1) activates polycystin-2 (PC2), resulting in calcium (Ca^2+^) influx. This generates a cascade of various protein activation and ultimately leads to endothelial nitric oxide synthase (eNOS) activation, producing vasodilator nitric oxide (NO). Calmodulin (CaM), calcium-dependent protein kinase (PKC) and serine-threonine kinase/protein kinase B (Akt/PKB) are involved in maintaining a healthy vascular structure. Abnormality in primary cilia has been proposed to promote vascular atherosclerotic formation.

**Table 1 cells-07-00233-t001:** Ciliary function and disease relevance.

Function	Disease Relevance	Reference
Nodal flow sensing	Situs inversus; Situs ambiguous; Situs isomerism	[14,15]
Mechano-sensing	Kidney, Liver, and Pancreas Diseases	[16,17,18]
Shear stress sensing	Hypertension; Atherosclerosis; Aneurysm formation	[10,11,12,13]
Osmolarity sensing	Respiratory diseases; Infertility	[19,20]
Gravitational sensing	Osteoporosis; Chondroporosis	[21,22,23]
Olfaction sensing	Anosmia; Hyposmia	[24,25]
Light sensing	Retinitis pigmentosa; Blindness	[26,27,28]
Chemo-sensing	Nephrocystin; Diabetes; Obesity	[29,30,31]
Neurotransmitter sensing	Impaired brain plasticity	[32]
Developmental regulatory sensing	Developmental defects; Cancer	[33,34,35]
Pressure sensing	Bone maintenance, development	[22,36,37]

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
