# Peer review of "The Roles of Primary Cilia in Cardiovascular Diseases"

_cells, 2018, doi:10.3390/cells7120233_

Round 1

Reviewer 1 Report

This review on' The Roles of Primary Cilia in Cardiovascular Diseases' by  

Rajasekharreddy Pala, Maha Jamal, Qamar Alshammari and Surya M. Nauli.

This review is interesting to the field but at the current state it is not ready to be published. I find it unbalanced because the introductory part needs more work in general.  

Some of the other sections are too shallow and need more detail required for a better review.

In particular,

line 73/91 : What do the authors mean by transition membrane? the correct name is transition zone. 

line 61: Where is the clear evidence that tubulin acetylation is for axonemal support?

line 64: where is the evidence that nodal cilia have radial spokes?

line 66: spelling actin reads action. What do you mean by assembly here?

line 98: not clear at all what the authors what to say. incomplete?

line 152: cardiac primary cilia function should not be confused with node cilia. Perhaps separate the 2 different developmental steps and where are these cilia. Do not forget that mutants for ciliary genes will affect both types of cilia: motile and immotile and therefore it is necessary to be critical when reading the results. 

line 161: too vague...

line 291 : The role of primary cilia in Atherosclerosis is a bit confusing.

line 316: Cilia in proliferation could be more discussed. For instance, cilia are present in some tumors but not in others. This diserves more discussion.

line 339 : Conclusions are sparce.

Author Response

Reviewer#1

Authors sincerely appreciate Reviewer’s concerns and suggestions.  We now have modified the manuscript using the Track Changes function from the MS Word, per the editorial request.  Below are the point-by-point responses.  Due to the Track Changes function, the line# has been slightly shifted.

line 73/91: What do the authors mean by transition membrane? the correct name is transition zone.

Transition membrane denotes lipid bilayer membrane that separates outside and inside a cell, whereas transition zone is generally referred to the machineries that are inside a cell including the transition fibers, ciliary necklace, basal body or centrosome.  Because ciliary-membrane and plasma-membrane have different lipid bilayer compositions [REF#53], transition-membrane connects ciliary and plasma membrane [REF#54,55].  This transition-membrane which is also known as periciliary-membrane forms a ciliary pocket.  We have now clarified this in the revision.

line 61: Where is the clear evidence that tubulin acetylation is for axonemal support?

Post-translational acetylation of tubulin is generally used as a marker of stable microtubules, such as in axonemes, mid-bodies, and mitotic spindles.  We have now included references relevant to the ciliary axoneme [REF#39-41].  Although acetylation is probably the most studied for ciliary axonemes, glutamylation and glycylation are also involved in post-translational modifications in the axonemes.  A recent study, for example, shows that an abnormal glycylation of ciliary tubulin prevents cilia formation and induces cancer formation [REF#176].

line 64: where is the evidence that nodal cilia have radial spokes?

The evidence of radial spokes in nodal cilia can be found in Ref#44

[https://www.cell.com/developmental-cell/fulltext/S1534-5807(15)00628-0].

line 66: spelling actin reads action. What do you mean by assembly here?

Thank you for point this out, and we have made the corrections for the typo and clarification accordingly.

line 98: not clear at all what the authors what to say. incomplete?

We have now clarified the sentence.

line 152: cardiac primary cilia function should not be confused with node cilia. Perhaps separate the 2 different developmental steps and where are these cilia. Do not forget that mutants for ciliary genes will affect both types of cilia: motile and immotile and therefore it is necessary to be critical when reading the results.

Thank you for your insightful comment.  We have now revised the section accordingly.

line 161: too vague...

We have explained the potential signaling pathways found in the literatures, including the Hh, TGFb and PDGF.

line 291: The role of primary cilia in Atherosclerosis is a bit confusing.

We apologize for the confusion.  The role of cilia in atherosclerosis is still controversial.  A lot more research is needed to clarify the confusion.  For example, on one hand increased ciliation is linked to atherosclerosis [REF#13], on the other hand cilia are protected against atherosclerosis [REF#162].

line 316: Cilia in proliferation could be more discussed. For instance, cilia are present in some tumors but not in others. This diserves more discussion.

Reviewer is correct that cilia are present in some tumors [REF#33,174,175] but not in others [REF#169-171].  The potential roles of cilia in tumors are still relatively new, and we have provided discussion on a potential posttranslational modification that is relevant to cilia and tumor formation.

line 339: Conclusions are sparce.

To avoid repetition as pointed by Reviewer#2, we have now added only one new paragraph stressing on the roles of cilia in cardiovascular disorders and the need of our continuous translational research toward a clinical intervention.

Reviewer 2 Report

This is a very readable and interesting article concerning cilia and the role in cardiovascular disease.

Some minor points and suggestions are below to help balance the opinion.

The article is quite repetitive in places - these should be edited out and shortened

line 31 such as cadiovasc diseases - probabaly not the most prominent cause - would refer to sever developmental defcts such as neural tube defects etc (eg Meckel syndrome)

line 37 list of HUMAN ciliopathies

Figure 1 - the primary cilia shows inner and outer dynein arms

line 94 define Hansens node

line 136 this is the most contentious part - I am sure the authors are aware of

Delling M, Indzhykulian AA, Liu X, Li Y, Xie T, Corey DP, Clapham DE. Primary cilia are not calcium-responsive mechanosensors. Nature. 2016 Mar 31;531(7596):656-60. doi: 10.1038/nature17426. Epub 2016 Mar 23. PubMed PMID: 27007841; PubMed Central PMCID: PMC4851444.

This paper should be discussed as it provides an alternative explanation to the cilia and flow hypothesis.

This topic (cilia bending and calcium influx) get mentioned a few more times

line 141 - discuss expression pattern of pkd1 and pkd2 - ie it is widely expressed across the vasculature

line 206 - typo abnormality rather than abnormal

The paper concentrates on pc-1 and pc-2 so this paper should be disussed:

Su Q, Hu F, Ge X, Lei J, Yu S, Wang T, Zhou Q, Mei C, Shi Y. Structure of the human PKD1-PKD2 complex. Science. 2018 Sep 7;361(6406). pii: eaat9819. doi: 10.1126/science.aat9819. Epub 2018 Aug 9. PubMed PMID: 30093605.

Author Response

Reviewer#2

Authors are grateful to many constructive suggestions from the reviewer.  Some repetitive statements have been modified or deleted.  Per the editorial guideline, all modifications in our revision are indicated in the Track Changes function of the MS Word.  The line# has therefore slightly been shifted, and the line-by-line responses to the reviewer’s comments are shown below.

line 31 such as cadiovasc diseases - probabaly not the most prominent cause - would refer to sever developmental defcts such as neural tube defects etc (eg Meckel syndrome)

Reviewer is correct that neural tube defects as seen in Meckel syndrome are probably more devastating than cardiovascular diseases.  We have now revised the paragraph accordingly.

line 37 list of HUMAN ciliopathies

We have now added the word “human”.

Figure 1 - the primary cilia shows inner and outer dynein arms

We have now replaced Figure 1, and thank you for pointing it out.

line 94 define Hansens node

We have now defined Hansens node as the site which determines the patterns the anterior-posterior axis of the embryo during gastrulation.

line 136 this is the most contentious part - I am sure the authors are aware of Delling M, Indzhykulian AA, Liu X, Li Y, Xie T, Corey DP, Clapham DE. Primary cilia are not calcium-responsive mechanosensors. Nature. 2016 Mar 31;531(7596):656-60. doi: 10.1038/nature17426. Epub 2016 Mar 23. PubMed PMID: 27007841; PubMed Central PMCID: PMC4851444.

This paper should be discussed as it provides an alternative explanation to the cilia and flow hypothesis. This topic (cilia bending and calcium influx) get mentioned a few more times

We appreciate the reviewer’s insight in this controversy. The Nature paper [REF#124] indeed argues that the presence of intraciliary Ca2+ (not Ca2+ cytosolic) signaling was not involved in mechanosensing function of cilia.  Many experts in the field have met and discussed this finding in many occasions.  The consensus is that Delling et al [REF#124] did not see calcium signaling in the cilium was possibly due to the baseline calcium fluorescence reaching 95% of saturation.  Please see graph below. 

Prior to Delling et al [REF#124], 4 independent laboratories have reported the mechanosensory role of ciliary calcium [PMIDs: 24056873, 24104765, 25660539, 26029358].  These laboratories observe changes in ciliary calcium by using G-GECO1.0, which has a Kd value of 749 nM.  On the other hand, Delling et al uses G-GECO1.2, which has a Kd value of 442 nM.  With this low Kd (442 vs. 749 nM), the baseline ciliary calcium signal would have reached saturation of the fluorescent signal intensity by about 95%.  Thus, any changes in ciliary calcium cannot be observed.

Note that we have discussed about this in our previous papers [PMIDs: 27341444, 29143784].  Although the ciliary calcium signaling is controversial, there is a consensus that cytosolic calcium is involved in mechanical fluid-shear stress.  Of importance is that it is generally recognized that primary cilia are mechanosensory organelles.  Per reviewer’s kind suggestion, we have added on the consensus of the cytosolic calcium in the revision.

line 141 - discuss expression pattern of pkd1 and pkd2 - ie it is widely expressed across the vasculature

Reviewer is correct that both pkd1 and pkd2 have long been recognized to be expressed in vasculature [REF#96-98], supporting a direct pathogenic rols of pkd1 and pkd2 in cardiovascular disorders.  We have briefly discussed this in the revision.

line 206 - typo abnormality rather than abnormal

Thank you for pointing this out, and it has been revised accordingly.

The paper concentrates on pc-1 and pc-2 so this paper should be disussed:

Su Q, Hu F, Ge X, Lei J, Yu S, Wang T, Zhou Q, Mei C, Shi Y. Structure of the human PKD1-PKD2 complex. Science. 2018 Sep 7;361(6406). pii: eaat9819. doi: 10.1126/science.aat9819. Epub 2018 Aug 9. PubMed PMID: 30093605.

We have now discussed the paper per your suggestion [REF#91].  Thank you.